# Manufacture-friendly nanostructured metals stabilized by dual-phase honeycomb shell

Hai Wang [1], Wei Song[1,2], Mingfeng Liu[2,3], Shuyuan Zhang[1], Ling Ren [1✉], Dong Qiu [4✉], Xing-Qiu Chen [3] & Ke Yang[1]

Refining grains to the nanoscale can greatly enhance the strength of metals. But the engineering applications of nanostructured metals are limited by their complex manufacturing technology and poor microstructural stability. Here we report a facile "Eutectoid element alloying→ Quenching→ Hot deformation" (EQD) strategy, which enables the mass production of a Ti6Al4V5Cu (wt.%) alloy with α-Ti grain size of 95 ± 32 nm. In addition, rapid co-precipitation of $Ti_2Cu$ and β phases forms a "dual-phase honeycomb shell" (DPHS) structure along the grain boundaries and effectively stabilizes the α-grains. The instability temperature of the nanostructured Ti6Al4V5Cu alloy reaches 973 K ($0.55T_m$). The room temperature tensile strength approaches 1.52 ± 0.03 GPa, which is 60% higher than the Ti6Al4V counterpart without sacrificing its ductility. Furthermore, the tensile elongation at 923 K exceeds 1000%. The aforementioned strategy paves a new pathway to develop manufacture-friendly nanostructured materials and it also has great potential for application in other alloy systems.

[1] Shi-changxu Innovation Center for Advanced Materials, Institute of Metal Research, Chinese Academy of Sciences, Shenyang, China. [2] School of Materials Science and Engineering, University of Science and Technology of China, Shenyang, China. [3] Shenyang National Laboratory for Materials Science, Institute of Metal Research, Chinese Academy of Sciences, Shenyang, China. [4] Centre for Additive Manufacturing, School of Engineering, RMIT University, Melbourne, VIC, Australia. ✉email: lren@imr.ac.cn; dong.qiu2@rmit.edu.au

Nanostructured metals (with grain size below 100 nm) exhibit ultrahigh strength and hardness, making them very attractive for developing novel lightweight and energy-efficient structural components[1–3]. However, the large volume fraction of grain boundaries provide a strong driving force for grain coarsening. In most nanostructured metals, prominent grain growth is observed in a temperature range of 0.25~0.4$T_m$ ($T_m$ is the melting temperature)[4]. Under loading conditions, the stability of nanostructured materials is further degraded. For example, in nanostructured pure Al and Ni[5,6], mechanically driven grain growth was observed during plastic deformation at room temperature. The inherent thermal and mechanical instability of nanostructured metals render them difficult to be manufactured into bulky components, which severely limit their application in engineering practice. Consequently, it is a long-term endeavor for material scientists to develop nanostructured metals with higher stability.

Over the past few decades, extensive investigations have shown that nanostructures can be stabilized through either thermodynamic or kinetic strategies[5,7–13]. Thermodynamically, lowering grain boundary energy can reduce the driving force for grain coarsening. This is often achieved by solute segregation, such as in Ni-W, Co-P, and Ni-Fe alloys[10,11,13]. In addition, using low angle boundary or twin boundary architectures in pure Cu or Ni can stabilize nanostructures as well[2,9,14]. Kinetically, the driving force for grain coarsening could be counteracted by precipitate particles pinning the grain boundaries. This is normally achieved by mechanical alloying, such as in Cu-WC and Cu-Ta alloys[5,12]. The stability of nanostructure can be further enhanced when thermodynamic and kinetic strategies are favorably combined together. However, this requires a more ingenious nanostructured design strategy that should not only employ low-energy interfaces, but also introduce thermally stable secondary phases pinning on the low-energy interfaces.

In this work, we have developed an idea to stabilize nanosized grains by a dual-phase honeycomb shell (DPHS) nanostructure in the Ti6Al4V5Cu model alloy (Fig. 1a). Here, equiaxed nanograins are analogous to the compartments in a honeycomb, they are fully encapsulated in thin dual-phase shells. The phases which make up of the dual-phase shells have low interfacial energy with the matrix. Hence, conventional high angle grain boundaries (HAGBs) with poor thermal stability are replaced by low-energy phase boundaries to thermodynamically stabilize the nanostructure. Moreover, the growth of any phase in the dual-phase shell is constrained by the other, thus the shell itself is of high stability. Such stable shells can exert an effective pinning force on nanograins to kinetically stabilize the nanostructure when it is exposed to high temperature and/or plastic deformation. The microstructural design concept, which synchronizes the thermodynamics and kinetics strategies prominently enhances the stability of the nanostructured metallic materials.

## Results and discussion

The microstructural thermal stability of the as-fabricated Ti6Al4V5Cu alloy with equiaxed α grains of 95 ± 32 nm (Supplementary Fig. 1) was determined by exposure for one hour at various temperatures (Fig. 1b). The onset instability temperature, was identified as high as 973 K (0.55$T_m$). This is notably higher than those of the nanostructured metals fabricated through conventional "bottom up" or "top down" technologies at the same grain size level (~100 nm)[4–7,15,16]. The mechanical properties of as-fabricated samples were tested at room temperature. For comparison, a commercial Ti6Al4V alloy consisting of equiaxed α grain with a size of 8.9 ± 3.8 μm (Supplementary Fig. 2) was also tested at the same loading conditions. The tensile strength of as-fabricated Ti6Al4V5Cu alloy is (1.52 ± 0.3) GPa, which is 60% higher than that of the Ti6Al4V counterpart without sacrificing its ductility (Supplementary Fig. 3). The comprehensive mechanical properties were greatly improved compared with commercial α, β, or (α + β) titanium alloys in the ASTM standard (Fig. 1c). The phenomenon of strength and ductility trade-off which was often reported in other nano-grained materials did not occur in the present study[14]. This can be attributed to the high mechanical stability inhibiting the occurrence of strain localization and early necking (Supplementary Fig. 4a). Furthermore, we performed high-temperature tensile tests at 923 K to examine the stability under a thermomechanical coupling condition (Fig. 1d and Supplementary Fig. 4b). Surprisingly, the model alloy was not fractured when the elongation exceeded 1000%, meanwhile, the fine grains still remained their initial size after the deformation. This superplasticity and high stability enable this nanostructured material to be easily manufactured into complex components through common bulk metal forming processes.

The high stability and mechanical properties originate from the alloy's unique microstructure. Transmission electron microscopy (TEM) analysis performed using high-angle annular dark field (HAADF) imaging indicates that alloying elements segregated along the nano-grain boundaries to form a shell structure (Fig. 2a). We randomly characterized a nano-grain by X-ray energy dispersive spectroscopy (XEDS) analysis (Fig. 2b). The equiaxed, aluminum enriched α-Ti nano-grain is fully enveloped by copper enriched $Ti_2Cu$ phase and vanadium enriched β phase. For these reasons, we have called it the dual-phase honeycomb shell (DPHS) structure.

The formation of nanosized α-Ti grains and DPHS structure is promoted by our 'Eutectoid element alloying→Quenching→Hot deformation' (EQD) strategy (Fig. 2c). Here, the E step is to alloy metals with eutectoid-forming elements that will contribute to the formation of eutectoid intermetallics constituting the shell in the last step (D step). The Q step refers to quenching the material at above the eutectoid transformation temperature. A combination of E and Q steps strongly suppress the eutectoid reaction, which will generate severe lattice distortion in martensite as a result contributing to the formation of a nanoscale lath precursor. The role of the D step is to convert the nanoscale lath precursor into an equiaxed nano-grained structure through hot deformation. Meanwhile, the dual-phase shell formed as a result of strain-assisted phase transformations. The EQD process can be achieved by traditional hot processing technologies that are compatible with the present industrial production lines (Supplementary Fig. 5). This circumvents the dilemma of high-cost and low-efficiency in the current nanostructured metal fabrication strategies[5,6,15,16]. For titanium alloys, eutectoid-forming elements include Cu, Si, Co, Ni, Mn, W, Cr, et. al. Given a high solubility of Cu in β titanium, a big atomic radius difference with respect to titanium, and the fact that $Ti_2Cu$ phase can rapidly precipitate from α titanium[17], Cu was selected as the additional alloying element in the most widely used Ti6Al4V alloy.

To determine the formation mechanism of the nanoscale lath precursor in E and Q steps, we performed X-ray diffraction (XRD) analysis on the as-quenched Ti6Al4V5Cu alloy (Fig. 2d). The specimen was mainly identified as α′ phase, hence the eutectoid reaction $β → α + Ti_2Cu$ was substantially inhibited. Given supersaturated copper in the α′ matrix (Supplementary Fig. 6), and large atomic radius difference between titanium (0.147 nm) and copper (0.128 nm), a large lattice strain should be stored in the martensite matrix. This lattice strain is reflected by the significant peak broadening in the XRD spectrum compared to the copper free Ti6Al4V specimen. Further calculation using the Williamson-Hall formula indicates that the average lattice strain increases from 0.186% to 0.228% due to copper alloying

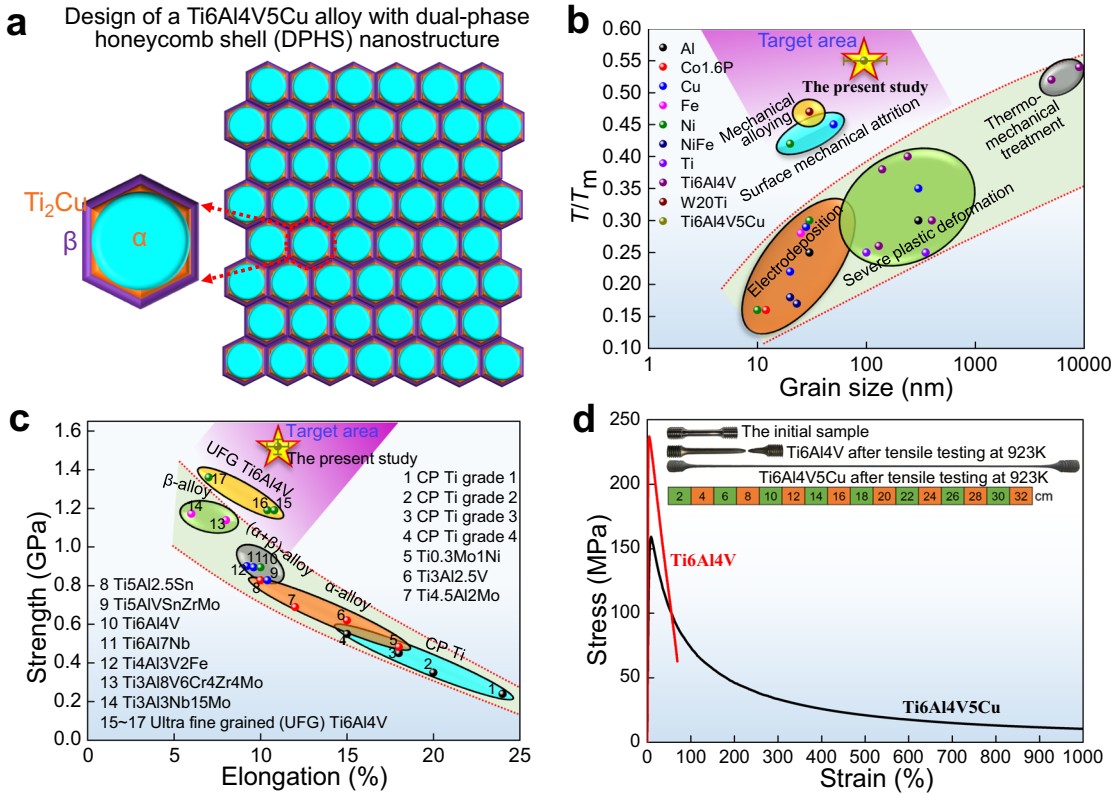

**Fig. 1 Design and properties of a nanostructured Ti6Al4V5Cu alloy with high stability. a** A dual-phase honeycomb shell (DPHS) nanostructure.
**b** Instability temperature versus average grain size. Literature data for metals processed with electrodeposition[14], severe plastic deformation[29–31], thermo-mechanical treatment[32,33], surface mechanical attrition[2,9], and mechanical alloying[8]. The error bar indicates the standard deviation of the measured grain size. **c** Tensile strength versus elongation. Literature data for titanium alloys in the ASTM standard and ultrafine grained Ti6Al4V alloys[15,16]. Error bars indicate standard deviations of the measured strength and elongation. **d** Stress versus strain curves at 923 K, showing the superplasticity of the Ti6Al4V5Cu alloy.

(Fig. 2e). It is well known that self-accommodation of lattice distortion is achieved by different lathy variants in neighbors[18]. To accommodate the excessive lattice distortion in the copper supersaturated matrix, higher number density of thin lath variants should be formed. This is consistent with the TEM observation that copper alloying resulted in a significant decrease in the lath width from 100–500 nm to 10–70 nm (Fig. 2f).

In order to convert the nanoscale lath precursor to an equiaxed nano-grained structure during the D step, transverse boundaries that are perpendicular to the longitudinal direction of lath martensite should be introduced. Experimental results showed that all the prismatic planes of hcp α phase are parallel with the longitudinal direction of laths (Fig. 2f and Supplementary Fig. 7). Therefore, only when the prismatic slip is the dominant deformation mode, transverse boundaries can be extensively generated in the material. This agrees with our TEM and electron back-scatter diffraction (EBSD) analysis (Supplementary Fig. 8). In the Ti6Al4V5Cu alloy, a strong prismatic texture that originated from the prismatic slip was identified (Fig. 2g), hence transverse boundaries formed (Supplementary Fig. 9b) and the nanoscale lath precursor was successfully converted to an equiaxed nano-grained structure. In the Ti6Al4V alloy, however, an obvious basal texture that originated from the basal slip was observed, this gave rise to the forming of longitudinal boundaries and thus the initial fine laths got coarsened during the deformation. It is generally accepted that the deformation mechanism in hcp metals is either basal or prismatic slip, depending on whether the c/a ratio is above or below 1.60[19]. The XRD measurement indicates that due to copper alloying, the c/a ratio decreased from 1.60 to

1.56 (Fig. 2d), thus explaining why prismatic slip dominated its hot deformation in the Ti6Al4V5Cu alloy.

While the newly formed equiaxed α′ nano-grains in the Ti6Al4V5Cu alloy have stable low angle grain boundaries (LAGBs) at the beginning of the hot deformation, would be expected to soon evolve into HAGBs through dynamic recovery and dynamic recrystallization (Supplementary Fig. 9c)[19]. From the perspective of lowering systematic energy, HAGBs tend to be spontaneously eliminated through grain coarsening[2]. That is why it is very challenging to preserve nanostructure after high-temperature deformation. However, in the current model alloy, strain-assisted martensite decomposition α′→α + β + Ti₂Cu was synchronously triggered by the hot deformation (Supplementary Figs. 9d, e and 10). As the nucleation rate along HAGBs is several orders of magnitude faster than that in the crystal lattice[20], the conjugated β and Ti₂Cu phases rapidly precipitate at sites of otherwise unstable HAGBs, forming "protective" dual-phase shells that envelop the nano-grains. EBSD observations reveal that these shells did not only inhibit grain coarsening during the high-temperature fabrication process, but also significantly enhanced the stability of the nanostructure against post-fabrication annealing (Fig. 3a). When annealed below 923 K for 1 h, both the nanostructure and the texture characteristic were barely evolved. That is why the onset thermal instability is determined as 973 K (Fig. 1b).

The high thermal stability we observed supports the belief that nanostructures can be remarkably stabilized when thermo-dynamic and kinetic strategies were favorable combined together through constructing the DPHS structure. Thermodynamically,

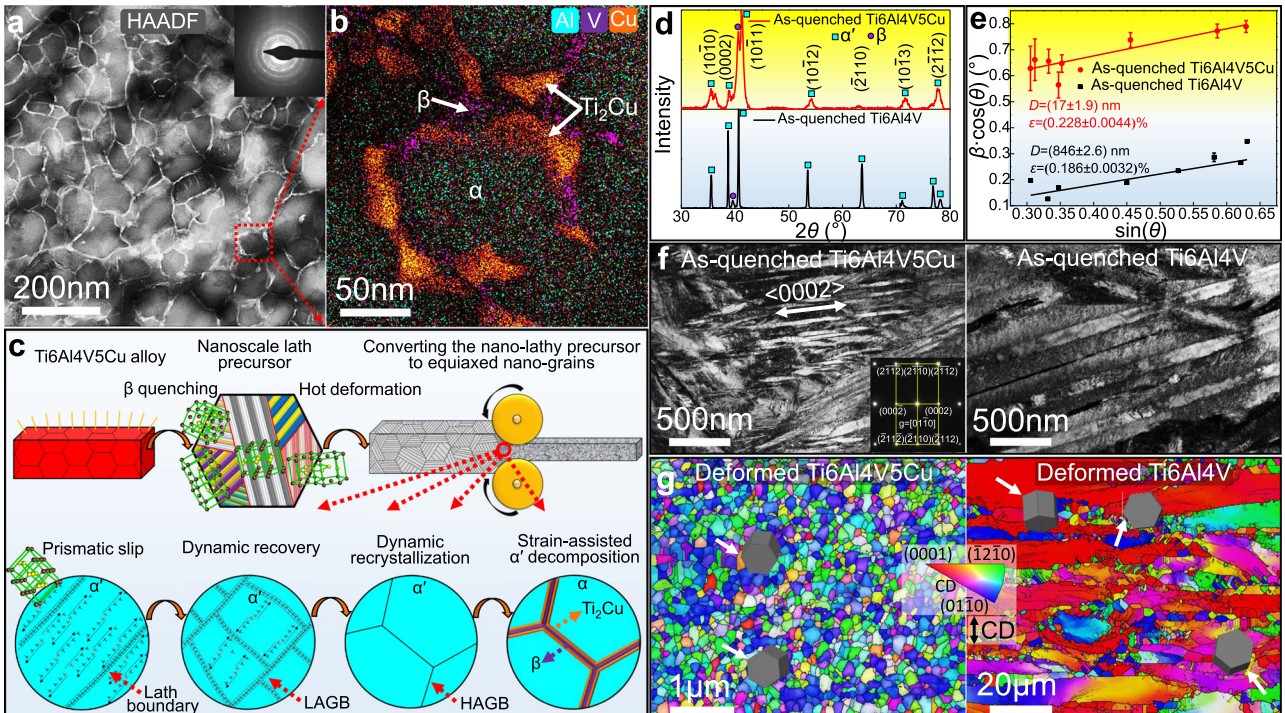

**Fig. 2 Microstructure and formation mechanisms. a** Transmission electron microscopy (TEM) performing under the high-angle annular dark field (HAADF) mode of the as-prepared Ti6Al4V5Cu alloy. **b** X-ray energy dispersive spectroscopy (XEDS) analysis. **c** Schematic showing the eutectoid element alloying, quenching and hot deformation (EQD) fabricating strategy. **d** X-ray diffraction analysis (XRD) of as-quenched materials. **e** $\beta \cdot \cos(\theta)$ versus $\sin(\theta)$ curves. Lattice strain $\varepsilon$ and crystalline domain size $D$ were determined by slopes and intercepts of curves. Error bars indicate standard deviations of measured $\beta \cdot \cos(\theta)$ value. **f** TEM bright field images of as-quenched materials. **g** Inverse pole figures (IPF) of deformed materials, compressive direction (CD) is denoted in the figure.

the driving force for grain coarsening was noticeably reduced when conventional high-energy HAGBs were replaced with low-energy α/Ti₂Cu or α/β phase interfaces. By tilting the specimens under high-resolution TEM (Fig. 3b), we found a set of low-lattice-misfit orientation relationships between the core α phase and the shell Ti₂Cu/β phase: $(0001)_\alpha//(013)_{Ti2Cu}//(10\bar{1})_\beta$; $[11\bar{2}0]_\alpha//[100]_{Ti2Cu}//[111]_\beta$. Based on the first-principles calculations (Fig. 3c, Supplementary Table 1), the phase interfacial energies of α/Ti₂Cu and α/β are only of 0.070 J/m² and 0.219 J/m² respectively, which are an order of magnitude smaller than that of conventional α/α HAGBs of 0.8–2.0 J/m²,[14]. This indicates that both types of interface enclosing the nanosized α grains are thermodynamically stable. It should be noted that the low-energy phase interface in the present study is different from the low-energy LAGBs which could gradually accumulate their misorientations at high temperatures to form high-energy HAGBs[21]. Our pole figure results (Fig. 3d) proved on a macro level that the initial low-lattice-misfit orientation relationship among α, Ti₂Cu and β phase was still maintained even after annealing at 973 K.

Kinetically, the Ti₂Cu/β dual-phase shells would exert a substantial pinning force on α grains and hence offsetting the driving force for grain coarsening. The role of the Ti₂Cu/β dual-phase shells to a large extent depends on two important factors: Firstly, whether they can precipitate immediately after the formation of nanosized α grains; secondly, the thermal stability of the Ti₂Cu/β shells at elevated temperatures. To elucidate these key issues, we used atom probe tomography (APT) analysis to provide an insight into the Ti₂Cu/β shells. In the as-fabricated sample (Fig. 3e), there is a prominent copper concentration gradient around the Ti₂Cu phase. This implies that the Ti₂Cu phase was rapidly precipitating from α grains during hot deformation. The average copper concentration in the α phase is (1.4 ± 0.2) at%,

thereby, it can be estimated that over 70% of copper atoms have migrated from the matrix to Ti₂Cu phase. We also noticed that vanadium was rejected when Ti₂Cu phase was growing during hot deformation, and this facilitated the nucleation of vanadium enriched β phase around the Ti₂Cu phase. This explains why the Ti₂Cu and β phase are frequently found side-by-side with a conjugated relationship. When the sample was annealed at 923 K for 1 h (Fig. 3f), there is no solute copper left in the matrix and this implies that any further coarsening of Ti₂Cu particles will expend other smaller Ti₂Cu particles surrounded. But the presence of adjacent β phase impeded any conceivable mass transfer of copper atoms from surrounding grains. In a similar fashion, the Ti₂Cu phase acted as an effective barrier to the mass transfer of vanadium from grain interiors. Consequently, further growth of either phase in the Ti₂Cu/β dual-phase shell is constrained by the other, and thus the shell possessed high stability. As a comparison, we have also prepared a nanostructured Ti-5Cu alloy only with Ti₂Cu particles pinning on α grain boundaries (Supplementary Fig. 11). The material presented inferior microstructural stability because the Ti₂Cu particles readily coarsen at high temperatures.

More importantly, the Ti₂Cu/β shells can also stabilize the nanostructure under thermomechanical coupling conditions, which enabled the material to present superplasticity at elevated temperatures. In order to understand the mechanism of superplasticity, we performed in-situ scanning electron microscopy (SEM) on a fiducial marked sample prepared by focused ion beam milling (Fig. 4a, b). After loading at 923 K to an average tensile strain of 40%, we neither observed any dislocation slip band, nor change in grain size or aspect ratio (Fig. 4c). This precluded the occurring of conventional deformation mechanisms such as dislocation slip or Coble diffusional creep in coarse-grained

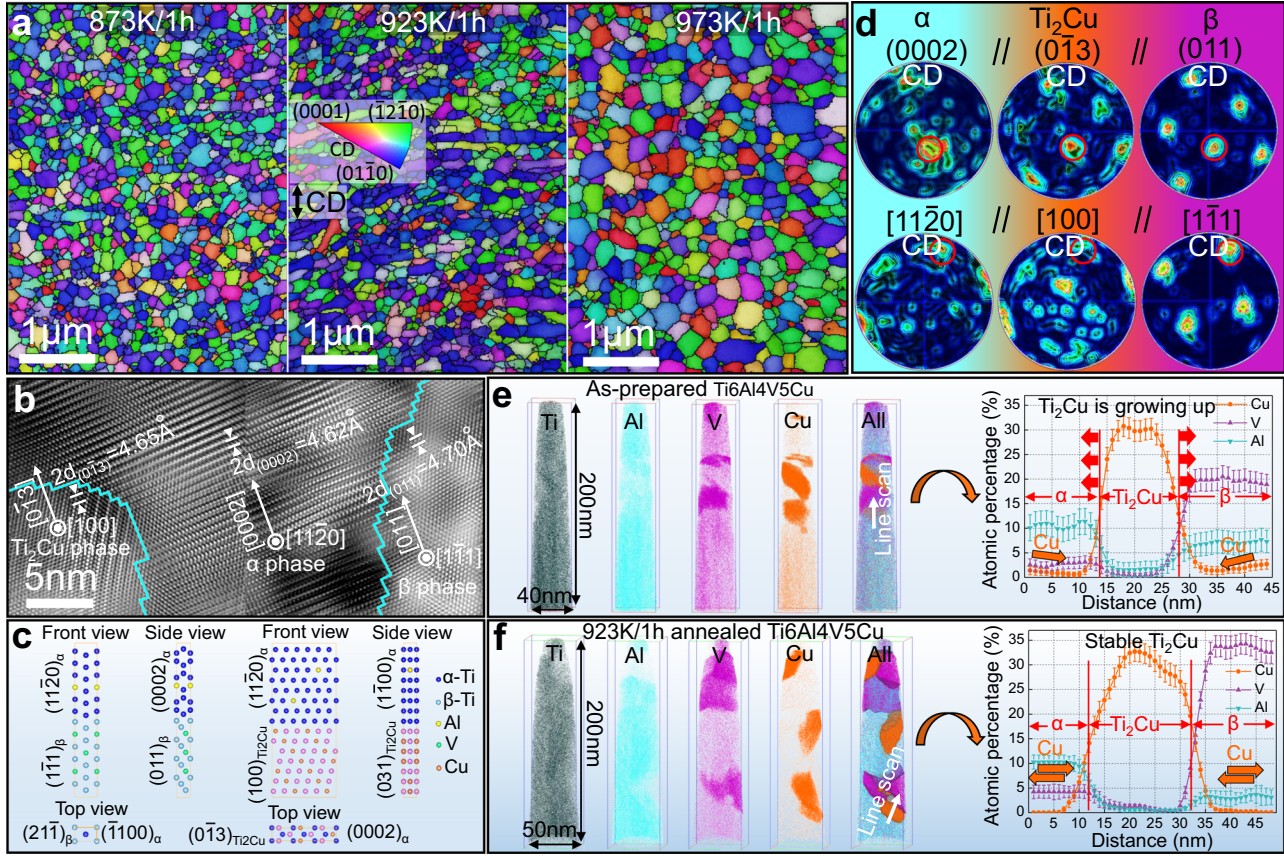

**Fig. 3 The DPHS nanostructure is stabilized thermodynamically and kinetically. a** IPF maps of the Ti6Al4V5Cu sample annealed at various temperatures, only the IPF maps of α phase is shown for brevity, compressive direction (CD) is denoted in the figure. **b** High-resolution TEM observation showing a low misfit orientation relationship of α/Ti₂Cu/β phases. **c** Interfacial structural models for first principles calculations. **d** Pole figures of the Ti6Al4V5Cu sample annealed at 973 K, the initial low-energy orientation relationship was retained. **e** Atom probe tomography (APT) analysis in the as-prepared Ti6Al4V5Cu sample, the copper concentration gradient indicating that the Ti₂Cu/β shell can be formed rapidly. **f** APT analysis of the 923 K/1 h annealed Ti6Al4V5Cu sample. Error bars in **e** and **f** indicate standard deviations of measured Al, V, Cu contents.

materials[22]. The fiducial markers were bent or even broken off at phase interfaces, demonstrating that interface sliding took place (Fig. 4d). We calculated the strain distribution in the material by measuring the displacement of the fiducial markers (Fig. 4e). Regions with larger strains always corresponded to the place where significant phase interface slip occurred, therefore, the superplastic deformation should be governed by phase boundary sliding, details of the superplastic mechanism has been shown in Fig. 4f. Although superplasticity has been reported in other nano-grained metals[23], it should be noted that the dominant deformation mechanisms in those studies were grain boundary migration and grain growth. As fine grains cannot be retained after the superplastic deformation, mechanical properties of the initial nanostructure are inevitably be reduced.

In this study, we have demonstrated a pathway to achieve ultra-stable nanosized grains by constructing a DPHS nanostructure through a low-cost EQD strategy in a Ti6Al4V5Cu model alloy. As a result, the key challenge to retain nanosized grains in metal bulk forming processes has been effectively overcome. Moreover, the EQD strategy is also applicable to other eutectoid alloy systems when a martensitic transformation is available. For example, we have also fabricated high stability Ti15Zr7Cu alloy with a DPHS nanostructure (Supplementary Figs. 12 and 13). We expect that the strategy reported here can be extended to other alloy systems, such as alloy steels, promoting the development and mass applications of bulky nanostructured metal products in future.

## Methods

**Materials.** For preparing the Ti6Al4V5Cu alloy, Ti, Al, V, Cu raw materials were consolidated using an anvil apparatus in which three orthogonal pistons compressed a cuboid chamber. Cuboid billets with the size of 180 × 180 × 450 mm were prepared then welded together forming a long billet. They were melted by vacuum consumable furnace to a 700 kg big ingot with the diameter of 380 mm. The ingot with the initial diameter of 380 mm was hot forged using a 16 MN hydraulic press machine at 1373 K, upset and drawn for three times with the reduction ratio of 3.0~4.0. Then it was forged into a round bar with the diameter of 110 mm (8 passes in total, reduction in diameter of each pass was <20%). Finally, the φ 110 mm round bars were forged into φ 60 mm round bars using a precision forging machine (6 passes in total, reduction in diameter of each pass was 8–12%) and the finish forging temperature was always kept above 1173 K. After hot forging, the bars were immediately quenched to room temperature in 30 wt.% NaCl water solution. Cylindrical samples with 60 mm in diameter and 140 mm in height were sectioned from the as-quenched forging bars. They were heated to 1013 K at a heating rate of 2–5 K/s, then held for 10 minutes and isothermally compressed by hydraulic press to a height of 15 mm under a compressive strain rate of 2 s⁻¹. The as-compressed Ti6Al4V5Cu samples were round plates with a diameter of (175 ± 10) mm. The upper and lower edges of the plate were removed with grinding machines by 2 mm. To clarify the formation mechanism of the DPHS nanostructure during hot deformation, additional hot compression tests at height reduction of 20%, 40%, 60%, 80% were also performed.

For preparing Ti6Al4V, Ti5Cu, and Ti15Zr7Cu alloys, their raw materials were consolidated to bulks through the anvil apparatus. Billets were melted using a 20 kg vacuum consumable furnace. The as-melted ingots were hot forged at above 1173 K into bars with the diameter of 60 mm and length of (0.8 ± 0.2) m, then immediately quenched to room temperature in a 30 wt.% NaCl water solution. Next, Ti6Al4V, Ti5Cu, and Ti15Zr7Cu alloys were compressed at 1013 K, 1033 K, and 983 K, respectively. Details of their fabricating procedures were the same as the aforementioned Ti6Al4V5Cu alloy. Chemical compositions (wt.%) of materials being studied were displayed in Supplementary Table 2.

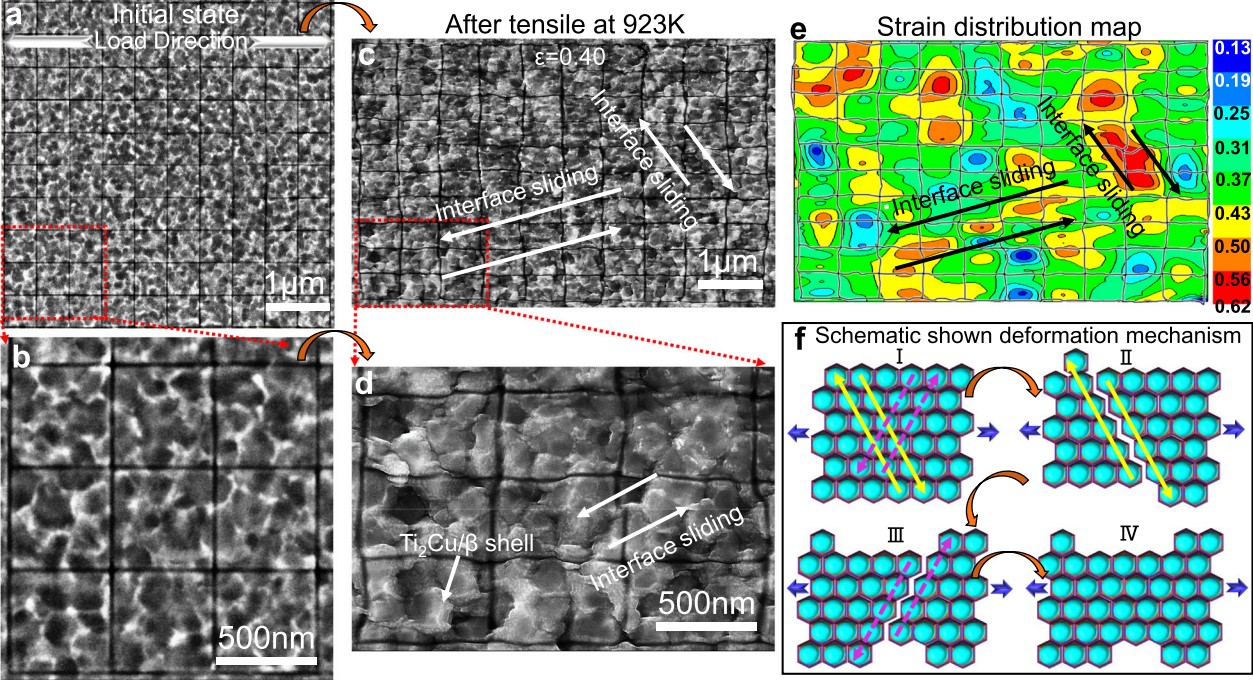

**Fig. 4 In-situ scanning electron microscopy (SEM) observation performing on a fiducial marked sample in the 923 K tensile test. a, b** SEM images of the sample before tensile test, shells of α grains can be viewed. **c, d** SEM images of the sample after tensile to the strain of 0.4, the bending of fiducial markers denotes phase interface sliding take place. **e** A strain distribution map, regions with larger strains correspond to where significant phase interface slip happens. **f** Schematic flowchart of the superplastic mechanism in the Ti6Al4V5Cu alloy: I, Initial state; II, Phase boundary sliding in one tangential direction (yellow arrows); III, Phase boundary sliding in another equivalent tangential direction (purple arrows); IV, Alternating phase boundary sliding in different tangential directions sustains superplastic deformation along the loading direction, while $Ti_2Cu/β$ shells suppress grain coarsening during the deformation.

**Tensile testing**. Tensile tests were performed on a Shimadzu AG-100KN universal testing machine which was equipped with a non-contact laser extensometer. Standard cylinder samples with the diameter of 5 mm and gauge length of 25 mm (ASTM standard E8/E8M-08) were used. They were machined with the use of the as-prepared Ti6Al4V5Cu plates. Tensile loading direction was perpendicular to the thickness direction of Ti6Al4V5Cu plates. Room temperature tensile tests were conducted with an initial strain rate of $0.001 \, s^{-1}$. High temperature tensile tests were conducted at 923 K with an initial strain rate of $0.01 \, s^{-1}$. The strength and elongation of samples were given by the averages of five parallel measurements.

**X-ray diffraction analysis**. Samples for XRD analysis were mechanically polished, then electrolytically polished using a mix solution of 5 vol.% perchloric acid, 35 vol.% n-butyl alcohol and 60 vol.% methanol at 25 V for 20 s. XRD experiments were performed on a Bruker D8 X-ray diffractometer using a Cu anode, the diffraction angular is in a range of 30° to 180° with a step size of 0.02°and a counting time of 5 s. The reflections of $\{10\bar{1}0\}$, $\{0002\}$, $\{10\bar{1}1\}$, $\{10\bar{1}2\}$, $\{2\bar{1}\bar{1}0\}$, $\{10\bar{1}3\}$ and $\{2\bar{1}\bar{1}2\}$ of the α′ phase were measured. The modified Williamson-Hall formula was used to calculate the lattice strain $ε$ and crystalline domain size $D$:[24]

$$\beta \cdot \cos(\theta) = \lambda/D + 5 \cdot \sin(\theta) \cdot \varepsilon \qquad (1)$$

where, $β$ is the integral peak width, $θ$ is the diffraction angle, $λ = 0.15418 \, nm$ is the X-ray wavelength.

**Electron backscatter diffraction**. Samples for EBSD analysis were sectioned along the compression direction by wire-electrode cutting, then mechanically grinded and polished, finally subjected to vibration polishing using a VibroMet2 machine in the nano silica polishing slurry for 24 h. The scanning step size of samples was 0.02 μm. Acquired data were analyzed using HKL-Channel 5 software after automatic noise reduction process. The color coding in the inverse pole figure (IPF), blue for $<10\bar{1}0>_α$, red for $<0001>_α$, and green for $<11\bar{2}0>_α$, gives the crystallographic orientation of each grain relative to the compressive direction (CD). Boundaries with misorientations larger than 15° were defined as high angle boundaries (HAGBs), they were labeled using bold lines. While boundaries with misorientations between 2 and 15° were defined as low angle grain boundaries (LAGBs), they were labeled using thin lines. Because β and $Ti_2Cu$ phases were covered by α grain boundaries, IPF maps cannot show the DPHS nanostructure. Pole figures of $<0001>_α$, $<11\bar{2}0>_α$, $<013>_{Ti2Cu}$, $<100>_{Ti2Cu}$, $<10\bar{1}>_β$, and $<111>_β$ were used to analyze the crystallographic orientation relationship between α, $Ti_2Cu$ and β phase. Schmid factor maps of the as-fabricated Ti6Al4V5Cu alloy were

calculated with a hypothesis that basal slip or prismatic slip dominated the hot deformation.

**Transmission electron microscopy**. Samples for TEM observation were ground to a thickness of below 40 μm. Foils with a diameter of 3 mm were punched out and two-jet thinning in an electrolyte of 5 vol.% perchloric acid, 35 vol.% n-butyl alcohol and 60 vol.% methanol at −30 °C using a voltage of 25 V, followed by ion-beam thinning at 2 keV using a Gatan PIPS™ facility. TEM observations were performed on a JEOL 2100 F microscope which was equipped with a X-ray energy dispersion spectrum detector and operated at 200 kV. High-angle annular dark field (HAADF) imaging was used to characterize the DPHS nanostructure. X-ray energy dispersive spectroscopy (XEDS) analysis was used to reveal the chemical composition distribution within the shell structure. High-resolution TEM imaging was used to identify α, $Ti_2Cu$ and β phase and to determine their crystallographic orientation relationship.

**Atom probe tomography**. Samples for APT analysis were fabricated using a dual-beam focused ion beam (FIB) on Helios Nanolab 600i from FEI. Rectangular cubes with dimensions of $2 \times 2 \times 10 \, μm$ were extracted from samples' surface then mounted on silicon microtips. They were subsequently trimmed by annular ion milling to sharp needles with a tip diameter of 30–60 nm and length of 6 μm. APT experiments were performed on a LEAP 4000X instrument, at a sample chamber temperature of 50 K, under a pulse frequency of 200 kHZ and a target evaporation rate of 0.8% per pulse. APT Data reconstruction was analyzed by Cameca IVAS 3.6 software.

**In-situ tensile scanning electron microscopy**. The specimen for the in-situ tensile test was a plate with gauge portion dimensions of 2 mm length × 3 mm width × 1 mm thickness, it was sectioned from the central part of the as-prepared Ti6Al4V5Cu plate. The surface was electropolished by a mix solution of 10 vol.% perchloric acid and 90 vol.% alcohol at −30 °C by a voltage of 25 V. In order to measure the strain distribution in the specimen, fiducial markers were milled on the surface using the FIB instrument. Markers were carved to the minimum possible depth to avoid cutting the sample. The dimensions of the fiducial markers were $5 \times 5 \, μm$ with pitches of 500 nm. The in-situ tensile experiment was performed on a micro-tensile testing stage, placed inside the chamber of a TESCAN MIRA high-resolution scanning electron microscope. The module is equipped with a resistance heater, which is capable of heating up to 1123 K. The temperature of the specimen was precisely controlled by a thermocouple wire spot welded on the

sample surface. The tensile test was conducted at 923 K at a strain rate of $0.01\ \text{s}^{-1}$. Every 10% of strain increment the tensile test was paused to acquire a SEM image.

**First-principles calculations of phase interface energies**. In order to calculate α/β and α/Ti₂Cu phase interface energies, we have constructed interfacial structural models based on the experimental orientation relationship of $(0002)_\alpha//(0\bar{1}3)_{\text{Ti2Cu}}//(011)_\beta$ and $[11\bar{2}0]_\alpha//[100]_{\text{Ti2Cu}}//[1\bar{1}1]_\beta$ (Fig. 3c). 24 atoms were used for building $(\bar{1}100)_\alpha-(2\bar{1}1)_\beta$ interface model and 90 atoms were used for building $(0002)_\alpha-(0\bar{1}3)_{\text{Ti2Cu}}$ interface model. For an interface model consisting of two phases M and N, the interfacial energy can be expressed as:

$$\gamma = \left( E_{sys} - mE_M^{bulk} - nE_N^{bulk} \right)/A - \sigma_M - \sigma_N \quad (2)$$

where $E_{sys}$ is the total energy of the interfacial modeling system, $E_M^{bulk}$ and $E_N^{bulk}$ are the energies in bulk M or N system in the unit of eV/atom, $A$ is the surface area of the interfacial structure, $m$ and $n$ are the number of atoms in the slabs of M and N. Here, $\sigma_M$ and $\sigma_N$ are the surface energy of the slab of M and N, which can be derived through the following equation as:

$$\sigma = \left( E_{slab} - nE_{bulk} \right)/2A \quad (3)$$

where $E_{slab}$ is the total energy of the relaxed slab, $E_{bulk}$ is the total energy of the bulk per atom, $A$ is the surface area of the slab. A 15 Å vacuum is used for the models to minimize the interaction in periodic images.

Using Density Functional Theory first principles calculations[25], we have calculated all the above energies by employing the Vienna ab initio simulation package (VASP)[26]. We have adopted the projected augmented wave (PAW)[27] method in describing the plane-wave basis and the generalized gradient approximation (GGA) of the Perdew-Burke-Ernzerhof (PBE) in describing the exchange-correlation functional[28]. The valence electrons of Ti, Cu, Al, and V atoms are treated as $3p^63d^24s^2$, $3d^{10}4s^1$, $3s^23p^1$, and $3p^63d^34s^2$. The cut-off energy of the plane waves is set at 400 eV. All the Brillouin zone integrations are performed on the Gamma centered $k$-mesh and sampled with a resolution of $2\pi \times 0.025\ \text{Å}^{-1}$.

## Data availability
The data generated in this study are provided in the Source Data file. Source data are provided with this paper.

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

## Acknowledgements
L.R. was supported by the National Key Research and Development Program of China (2018YFC1106600), LiaoNing Revitalization Talents Program (XLYC1807069). K.Y. was supported by the National Key Research and Development Program of China (2016YFC1100600) and National Natural Science Foundation (51631009). H.W. was supported by the Doctoral Scientific Research Foundation of LiaoNing Province (2020BS002). X.Q.C. was supported by National Natural Science Foundation (31870954 and 51725103).

## Author contributions
H.W. performed the experimental studies. W.S. and S.Z. carried out the analysis. M.L. performed modeling and simulation. D.Q. and X.C. gave guidance on experimental design. L.R. and K.Y. supervised the study.

## Competing interests
The authors declare no competing interests.
