## [Peer Review File · Nature Communications]

Reviewer #1 (Remarks to the Author):

Dear Editor and Authors,

I have completed my review of the manuscript by Wang et al. titled "Manufacture-friendly nanostructured metals stabilized by dual-phase honeycomb shell". In this work, the authors present a novel (to the best of my knowledge) methodology for designing and manufacturing alloys with sub-micron grain sizes, and successfully demonstrate their approach by manufacturing a novel titanium alloy with some unique properties. Specifically, the highly refined sub micron grain structure of the alloys appears to be very stable and allows production of material at bulk industrial scales without deterioration of the microstructure. The authors do demonstrate scalability by producing 700kg of material. The alloy produced using the authors' approach has excellent strength characteristics when compared to the widely used titanium alloy Ti6Al4V and also has outstanding superplasticity at elevated temperature. The authors have documented their work very well and have used all the modern materials characterisation techniques to study their material and to determine the operating mechanism for the unique behaviour. Both the new alloy and the manufacturing method can make a significant contribution to the field of metallurgy and materials science and I am therefore happy to recommend the manuscript for publication. Having said that, the authors would need to address the following points in a revised manuscript:

- 1). The authors do not discuss the aspect ratio of their nano grains. Are they equiaxed or elongated? The authors need to comment on this briefly in the text and provide a micrograph of a perpendicular section to supplementary figure 1.
- 2). The authors compare the tensile behaviour of their alloy and standard Ti6Al4V. The authors need to comment on the grain size (and alpha lath size) of the Ti-6-4 alloy tested and the microstructure it had. Ideally this should be done both in the text and for supplementary figure 2.
- 3). A bit more information is needed on the forging of the 60mm round bars from the ingot. If the authors used cogging as their ingot-to-billet conversion process, the details should be provided - no of passes, reduction, etc.
- 4). Supplementary figure 4 should have the isothermal compression stage explained better for the final 175mm plate specimen that was produced. In addition, under the picture of the plate there is a strange blue bar saying "never compromise safety" upside down. Is it a necessary element? It does not seem to fit. It may be the side view of the plate, but size does not match the top view and it looks too different.
- 5). The authors need to provide their VASP input files, supercell files and output data as supplementary materials.
- 6). Was aluminium ignored for the VASP calculations? The difference in electron structure from transition metals can make calculations tricky. The authors need to state clearly their assumption and give a justification/explanation as to why the aluminium would not affect the surface energy calculations in a significant way.

7). How many atoms were used for the DFT simulations?

8). The authors state that prismatic slip in the alpha phase is an important part of the deformation mechanism. The main experimental evidence for this is the prismatic texture of the EBSD maps. Ideally, the authors should also provide some TEM images showing abundant prismatic dislocations. Alternatively, some suitable references relating TEM observations to EBSD should be sufficient.

9). In figure 3c a legend explaining atom colours should be included.

10). The stages of deformation I, II, III and IV shown in figure 4f do not seem to be discussed explicitly anywhere in the the text, figure captions or supplementary material. This needs to be addressed.

11).The standard of English and quality of writing are acceptable, although there are some rough edges. I recommend making the following stylistic changes:

everywhere in the text - replace "thermal-mechanical" with "thermomechanical"

everywhere in the text - replace "nano-lathy precursor" with something better. It is not quite English, certainly not proper scientific English.

everywhere in the text - What are transversal boundaries? DO the authors mean transverse boundaries?

page 1, line 16 - change "be applied" to "apply"

page 1, line 21 - change "Ti6Al4V5Cu" to "Ti6Al4V5Cu (wt.%)"

page 1, line 30 - change "has a great" to "has great"

page 1, line 30/31 - change "to be applied in" to "for application to"

page 2, line 44 - change "Over past" to "Over the past"

page 2, line 51 - delete "Basing on above theories", start sentence with "The stability"

page 2, line 63 - change "exert effective" to "exert an effective"

page 2, line 65 - change "that synchronises the thermodynamics and kinetics strategies" to ", which synchronises the thermodynamics and kinetics strategies,"

page 2, line 66 - change "alloy as a model alloy in" to "model alloy for"

page 2, line 73 - change "by room temperature" to "using room temperature"

page 2, line 82 - change "remained its" to "retained their"

page 2, line 84 - change "manufactured complex" to "manufactured into complex"

page 3, line 95 - change "should be originated from its" to "originate from the alloy's"

page 3, line 96 - change "microstructural architecture" to "microstructure" (no need to be too fancy)

page 3, line 96/97 - change "under the high-angle ... mode" to "using high-angle ... imaging"

page 3, line 98 - change "forming" to "form"

page 3, line 101 - change "This is why we call it" to "For these reasons, we have called it the"

page 3, line 102 - change "is credited to" to "is promoted by" or "is enabled by"

page 3, line 108 - change "The D step" to "The role of the D"

page 3, line 109 - change "Meanwhile, dual-phase" to "Meanwhile, the dual-phase"

page 3, line 122 - change "It well" to "This"

page 4, line 116 - change "short-listed as the" to "selected as the additional"

page 4, line 117 - delete "in this study"

page 4, line 135 - change "Williamson-Hall" to "the Williamson-Hall"

page 4, line 145 - change "dominated" to "dominant"

page 5, line 156 - change "The newly" to "While the newly"

page 5, line 156 - change "might be with" to "have"

page 5, line 157 - change "but they would" to "would be expected to"

page 5, line 164 - change "phase rapidly precipitated" to "phases rapidly precipitate"

page 5, line 165 - change "shells enveloping" to "shells that envelop"

page 5, line 167 - change "nanostructured stability" to "stability of the nanostructure"

page 5, line 177 - change "energy" to "energies"

page 5, line 179 - change "phase interfaces" to "types of interface"

page 5, line 185 - change "exert substantial" to "exert a substantial"

page 5, line 185 - change "grains hence" to "grains and hence"

page 5, line 186 - delete "Obviously" (dangerous word to use in scientific writing)

page 5, line 189 - change "we use atom probe tomography (APT) analysis" to "we used atom probe tomography (APT)"

page 6, line 194 - change "transferred" to "migrated" or "diffused"

page 6, line 201 - change "to hamper" to "to"

page 6, line 202 - change "grain inside" to "grain interiors"

page 6, line 203 - change "thus" to "and thus"

page 6, line 205 - change "due to" to "because the"

page 6, line 206 - change "easily get coarsened" to "readily coarsen"

page 7, line 218 - delete "a" before therm...

page 7, line 219 - change "conditions" to "condition"

page 7, line 221 - delete "observation"

page 7, line 226 - change "phase interface" to "interface"

page 7, line 230 - change "phenomenon has ever" to "has"

page 7, line 231 - change "dominated" to "dominant"

page 7, line 233 - change "would" to "are"

page 7, line 246 - change "applicable in" to "applicable to"

page 8, line 253 - delete "to bulks"

page 8, line 254 - change "through" to "using"

page 8, line 269 - change "as aforementioned" to "as the aforementioned"

page 8, line 282 - change "electrolytic polished by" a mix" to "electrolytically polished using"

page 9, line 291 - change "backscattered" to "backscatter"

page 9, line 291 - delete "analysis"

page 9, line 295 - change "data was analysed by" to "data were analysed using"
page 9, line 295 - change "code" to "coding"
page 9, line 297 - change "pointing in the direction parallel" to "relative"
page 9, line 299 - change "in bold lines" to "using bold lines"
page 9, line 300 - change "in thin lines" to "using thin lines"
page 9, line 303 - delete "observation"
page 9, line 306 - change "by a voltage" to "using a voltage"
page 9, line 309 - change "mode were" to "imaging was"
page 9, line 310 - change "analysis were" to "analysis was"
page 9, line 311 - change "in the shell" to "within the shell"
page 9, line 311 - change "mode was" to "imaging was"
page 9, line 313 - delete "analysis"
page 9, line 316 - change "Following, they were" to "They were subsequently"
page 9, line 321 - delete "observation"
page 9, line 323 - change "thick" to "thickness"
page 9, line 325 - change "drawn" to "milled"
page 9, line 325 - change "by the FIB" to "using the FIB"
page 9, line 327 - change "Markers were carved as light as possible" to "Markers were carved to the minimum possible depth"
page 9, line 327 - change "Dimensions" to "The dimensions"
page 9, line 327 - change "markers was" to "markers were"

page 10, line 337 - change "In terms of" to "For an"
page 10, line 349 - change to "Using Density Functional Theory first principles calculations, we have ..."

Reviewer #2 (Remarks to the Author):

This manuscript reports a novel alloying and microstructural design strategy that enables mass production of a unique dual-phase honeycomb shell nanostructure in a Ti6Al4V5Cu alloy by industrial scale thermomechanical processing. The unique nanostructure is characterized by equiaxed alpha grains less than 100 nm in size with a dual-phase thin layer around the alpha grains. Such a nanostructure shows a high thermal and mechanical stability and a good combination of strength and ductility at room temperature and superplasticity at high temperature. The processing is simple and the results are interesting. The proposed strategy has been successfully applied to the alloy design and production of a similar nanostructure in a Ti15Zr7Cu alloy. The manuscript merits publication but the following comments have to be addressed before its acceptance.

A major concern is about the explanation for the formation of the dual-phase honeycomb shell nanostructure in the designed Ti6Al4V5Cu alloy during hot deformation (Fig. 3). The results showed that the addition of 5%Cu in a conventional Ti6Al4V alloy has enabled the formation of finer martensitic laths after quenching from the beta phase and changed the deformation mechanisms and microstructural evolution during subsequent hot compression at 1023 K. It is no doubt that the development of the nanoscale alpha grains with the dual-phase shell structure during the hot compression is related to the

combined effect of dynamic recovery and recrystallization with strain assisted phase transformation from alpha prime into alpha +beta+Ti₂Cu. However, the authors speculated a three-step process for the microstructural evolution during hot compression: (1) Transverse boundaries that are perpendicular to the lath boundaries are introduced by the activation of prismatic  slip, resulting in the formation of equiaxed alpha prime grains with stable low angle boundaries at the beginning of hot deformation; (2) the low angle boundaries soon evolve into high angle boundaries through dynamic recovery and dynamic recrystallization; (3) conjugated β and Ti₂Cu phases rapidly precipitate at the HAGBs, forming “protective” dual-phase shells enveloping the nano-grains. In the section of Methods, the heating rate to heat the sample to the hot compression temperature and the isothermal holding time before compression were not described. Therefore, it is not known whether the transformation of alpha prime to alpha phase already took place before compression. More importantly, no observation of the microstructure at lower compression strains was provided to support the speculation about the evolution from low angle to high angle grain boundaries before the precipitation of beta and Ti₂Cu at grain boundaries. A detailed electron microscopy study is needed before a conclusion can be reached about the formation mechanism of the dual-phase honeycomb shell nanostructure. In this manuscript, if further evidence cannot be provided, it is suggested to simplify the related description and discussion and leave a more detailed electron microscopy investigation to reveal the formation mechanism for a future study.

Fig. 3d shows pole figures of alpha, beta and Ti₂Cu phases. This indicates that the EBSD applied in this work allowed the detection of EBSD patterns from the thin layers of individual beta and Ti₂Cu phases. However, the EBSD orientation maps shown in Fig. 3a (also Fig. 2g) show only alpha grains without a shell structure, as revealed in the TEM images (Fig. 2a). How have the EBSD maps in Fig. 3a and Fig. 2g been processed? Have any data cleaning procedures been applied? If so, the EBSD data processing needs to be described in the Methods. Also, the sample coordinate system should be indicated in the pole figures in Fig. 3d.

In situ SEM observations have shown clearly that phase boundary sliding associated with the fine structure is the dominant deformation mechanism responsible for the excellent superplasticity. Microstructural observation in the sample deformed to 40% showed no grain growth. However, microstructural observations at higher strains should be presented to verify whether grain growth has occurred during the superplastic deformation. In conventional fine-grained superplastic materials, it has been well-established that gradual grain growth occurs during superplastic deformation, which is a mechanism for hardening that is considered to be helpful in addition to the strain rate hardening to maintain a smooth necking of the tensile specimen during superplastic deformation. The grain growth and its role in superplastic deformation in the present alloy cannot be ruled out before experimental evidence is provided.

At last but not least, there are quite a few grammatical and typographic errors in the main text and in figure captions. A careful check of the language is required. Examples of errors are:

Line 92: Strength versus stain curves (Stress versus strain curves)

Line 122 and Fig. 4f: Schematic shown..... (Schematic showing.....)

Line 138: lathy variants (lath variants)

Line 260: rolling bars (forging bars)

Lines 278 and 279: strain rate (strain rate)

Line 293: vibration polished (vibration polishing)

.....

** See Nature Research's author and referees' website at www.nature.com/authors for information about policies, services and author benefits.

Point by point response to reviewer comments

The authors sincerely appreciate reviewers' pertinent comments on our manuscript entitled "Manufacture-friendly nanostructured metals stabilized by dual-phase honeycomb shell" (NCOMMS-21-19879). These comments are highly valuable to improve the quality of our manuscript. The authors have completed all the additional experiments suggested by the reviewers and made a thorough revision. Amended parts have been highlighted in the revised manuscript. The point-by-point responses to reviewers' comments are as follows:

Reviewer #1

(1) The authors do not discuss the aspect ratio of their nano grains. Are they equiaxed or elongated? The authors need to comment on this briefly in the text and provide a micrograph of a perpendicular section to supplementary figure 1.

Response:

Thanks for your suggestions. We confirm that the as-fabricated nanosized grains are equiaxed as shown in Fig. 2a, 2b, 2g, 3a, 4a, 4b and Supplementary Fig. 1. According to your suggestion, we have also added some descriptions in the revised manuscript. Meanwhile, we have included a pair of additional micrographs on the vertical cross-section, as shown in the Supplementary Fig. 1b (OM) and 1e (TEM) below. The size and aspect ratio of the nanosized grains are very similar to those on the horizontal cross-section (Supplementary Fig. 1(c, d) and 1(f, g)). Accordingly, we have made the following amendments in the revised manuscript.

Page 2, line 68: "with an average grain size of" has been replaced by "with equiaxed α grains of."

Page 13, line 472: the Supplementary Fig. 1 has been updated as shown below.

Supplementary Fig. 1 As-fabricated nanostructured Ti6Al4V5Cu alloy. **a**, Sample profile overview. **b**, **c** and **d**, Optical micrographs at selected areas. **e**, **f** and **g**, TEM images of the microstructure in areas **b**, **c** and **d**, respectively.

(2) The authors compare the tensile behavior of their alloy and standard Ti6Al4V. The authors need to comment on the grain size (and alpha lath size) of the Ti-6-4 alloy tested and the microstructure it had. Ideally this should be done both in the text and for supplementary figure 2.

Response:

Thanks for your comments. According to your suggestions we have carefully characterized the microstructure of the Ti-6Al-4V alloy tested in this study using EBSD and TEM. Results show that the microstructure of this alloy consists of equiaxed α grains with the grain size of $8.9 \pm 3.8 \mu\text{m}$. Both the main text and the supplementary Fig. 2 were amended accordingly as follows.

Page 2, line 72: “We evaluated the mechanical properties of as-fabricated samples by room temperature tensile tests (Supplementary Fig. 2). The tensile strength is (1.52 ± 0.3) GPa and the elongation is $(11 \pm 1)\%$.” has been replaced by “The mechanical properties of as-fabricated samples were tested at room temperature. For comparison, a commercial Ti6Al4V alloy consisting of equiaxed α grain with a size of $8.9 \pm 3.8 \mu\text{m}$ (Supplementary Fig. 2) was also tested at the same

loading conditions. The tensile strength of as-fabricated Ti6Al4V5Cu alloy is (1.52 ± 0.3) GPa, which is 60% higher than that of the Ti6Al4V counterpart without sacrificing its ductility (Supplementary Fig. 3).”

Page 14, line 501- a new Supplementary Figure. 2 has been included. The ordinal number of other supplementary figures have been updated accordingly.

Supplementary Fig. 2 The microstructure of commercial Ti6Al4V base alloy consisting of equiaxed α grains. **a**, EBSD phase distribution map and **b**, α phase IPF map. **c**, Bright field TEM image close to an α grain boundary and **d**, TEM-EDS map of the area of **c**.

(3) A bit more information is needed on the forging of the 60mm round bars from the ingot. If the authors used cogging as their ingot-to-billet conversion process, the details should be provided – no. of passes, reduction, etc.

Response:

The initial ingot with a diameter of 380 mm was annealed in an argon gas purged furnace at 1373 K for 4 hours, followed by repetitive upsetting and drawing for three times using a 16 MN

hydraulic press machine with the reduction ratio between 3.0 and 4.0. Intermediate annealing at 1373K for 2 hours was operated, once the ingot was cooled below 1173 K during upsetting and drawing. Then the ingot was subjected to eight passes of multi-directional cogging to be shaped into a round bar with a diameter of 110 mm. The reduction in diameter of each pass was no more than 20%. Finally, the ϕ 110 mm round bars were forged into ϕ 60 mm round bars using a precision forging machine (as shown in the Supplementary Fig. 4VJ). In this last step, the round bars were subjected to six passes forging where the reduction in diameter of each pass was between 8% and 12% and the forging temperature was kept above 1173 K.

Basing on your comment, we have appropriately simplified the above details of forging process and supplemented them in the revised manuscript. Please find below the changes we have made:

Page 8, line 257-change “The ingot was hot forged at above 1173 K into bars with the diameter of (60 ± 2) mm and length of (1.5 ± 0.3) m.” to “The ingot with the initial diameter of 380 mm was hot forged using a 16 MN hydraulic press machine at 1373 K, upset and drawn for three times with the reduction ratio of 3.0~4.0. Then it was forged into a round bar with the diameter of 110 mm (8 passes in total, reduction in diameter of each pass was less than 20%). Finally, the ϕ 110 mm round bars were forged into ϕ 60 mm round bars using a precision forging machine (6 passes in total, reduction in diameter of each pass was 8~12%) and the forging-ending temperature was always kept above 1173 K.”

(4) Supplementary figure 4 should have the isothermal compression stage explained better for the final 175 mm plate specimen that was produced. In addition, under the picture of the plate there is a strange blue bar saying "never compromise safety" upside down. Is it a necessary element? It does not seem to fit. It may be the side view if the plate, but size does not match the top view and it looks too different.

Response:

We have fully adopted your suggestion and provided a photograph of isothermal compression as the last step of the fabrication procedures in the Supplementary Fig. 4VI. Meanwhile, we have deleted the original photograph of the ruler with irrelevant information. Details of changes can be seen below:

Page 16, line 561- change the supplementary Fig. 4.

Supplementary Fig. 4 Fabrication procedures for the nanostructured Ti6Al4V5Cu alloy

(5)The authors need to provide their VASP input files, supercell files and output data as supplementary materials.

Response:

Thanks for your suggestions. We have included all the VASP input files and supercell files, including INCAR, KPOINTS, POSCAR in the supplementary materials. Meanwhile, we have summarized our output data in a new Supplementary Table 1. Details of changes can be seen below:

Page 5, line 177-change “(Fig. 3c)” to “(Fig. 3c, Supplementary Table1)”

Page 8, line 271-change “Supplementary Table1” to “Supplementary Table2”

Page 23, line 765-add a new Supplementary Table 1.

Page 23 line 765-change “Supplementary Table1” to “Supplementary Table2”

Supplementary Table1 VASP output data

Surface energy calculation results

1a

Structure model	Interface Area (\AA^2)	Bulk energy (eV)	Slab energy (eV)	Surface energy ($\text{eV}/\text{\AA}^2$)
$(\bar{1}100)_\alpha$	13.099	-89.772	-86.433	0.127
$(2\bar{1}\bar{1})_\beta$	13.099	-95.243	-92.309	0.112
$(0002)_\alpha$	42.639	-365.395	-355.044	0.121
$(0\bar{1}3)_{\text{Ti}_2\text{Cu}}$	42.639	-273.133	-263.790	0.110

Structure model	Interface Area (\AA^2)	Model energy (eV)	Interfacial energy	
			($\text{eV}/\text{\AA}^2$)	(J/m^2)
$(\bar{1}100)_\alpha$ - $(21\bar{1})_\beta$	13.099	-181.699	0.0137	0.219
$(0002)_\alpha$ - $(0\bar{1}3)_{\text{Ti}_2\text{Cu}}$	42.639	-628.493	0.0044	0.070

(6) Was aluminum ignored for the VASP calculations? The difference in electron structure from transition metals can make calculations tricky. The authors need to state clearly their assumption and give a justification/explanation as to why the aluminum would not affect the surface energy calculations in a significant way.

Response:

We agree with the reviewer that aluminum should not be simply ignored in VASP calculations as we did in the original work. In the revised manuscript, we have built a new interface model considering the effect of aluminum (Fig. 3c). Based on this new model, the interfacial energies were recalculated, and a result comparison between with aluminum and without aluminum have been listed in Table R1. When considering the effect of aluminum, the phase interface energies of $\alpha/\text{Ti}_2\text{Cu}$ and α/β are $0.070 \text{ J}/\text{m}^2$ and $0.219 \text{ J}/\text{m}^2$ respectively, which are slightly higher than the case of without considering aluminum. Nevertheless, these values are still an order of magnitude smaller than that of conventional α/α HAGBs of $0.8\sim 2.0 \text{ J}/\text{m}^2$.^{2, 16} Hence, the amended calculation results still support that the *DPHS* nanostructure is thermodynamically stable.

Table R1 VASP calculation results considering Al and without considering Al

Structure model	Interfacial energy (J/m^2)	
	Without Al	With Al
$(\bar{1}100)_\alpha$ - $(21\bar{1})_\beta$	0.140	0.219
$(0002)_\alpha$ - $(0\bar{1}3)_{\text{Ti}_2\text{Cu}}$	0.044	0.070

In the revised manuscript, we have updated the VASP calculation results and the Methods section. Details of changes are as follows:

Page 5, line 177-change “the phase interfacial energies of $\alpha/\text{Ti}_2\text{Cu}$ and α/β are only of 0.044 J/m^2 and 0.140 J/m^2 ” to “the phase interfacial energies of $\alpha/\text{Ti}_2\text{Cu}$ and of α/β are only 0.070 J/m^2 and 0.219 J/m^2 , respectively”

Page 6, line 208-update the Fig. 3c with a new model considering aluminum.

Fig. 3c Interfacial structural models for first principles calculations

Page 10, line 346-delete “We have constructed surface models in which two topmost atomic layers are allowed to be relaxed to simulate the free surface and the other layers are fixed to describe bulk phase.”

Page 10, line 353-change “Ti, Cu and V atoms” to “Ti, Cu, Al and V atoms”

Page 10, line 354-change “ $3p^63d^24s^2$, $3d^{10}4s^1$ and $3p^63d^34s^2$ ” to “ $3p^63d^24s^2$, $3d^{10}4s^1$, $3s^23p^1$ and $3p^63d^34s^2$ ”

(7) How many atoms were used for the DFT simulations?

Response:

We have added the quantity of atoms for the DFT simulations in the revised manuscript. Details of changes are as follows:

Page 10, line 337- add “24 atoms were used for building $(\bar{1}100)_\alpha-(21\bar{1})_\beta$ interface model and 90 atoms were used for building $(0002)_\alpha-(0\bar{1}3)_{\text{Ti}_2\text{Cu}}$ interface model” after “we have constructed interfacial structural models basing on the experimental orientation relationship of $(0002)_\alpha//(\bar{0}\bar{1}3)_{\text{Ti}_2\text{Cu}}//(\bar{0}11)_\beta$ and $[11\bar{2}0]_\alpha//[100]_{\text{Ti}_2\text{Cu}}//[1\bar{1}1]_\beta$ (Fig. 3c).”

(8) The authors state that prismatic slip in the alpha phase is an important part of the deformation mechanism. The main experimental evidence for this is the prismatic texture of the EBSD maps. Ideally, the authors should also provide some TEM images showing

abundant prismatic dislocations. Alternatively, some suitable references relating TEM observations to EBSD should be sufficient.

Response:

We have followed the reviewer's suggestions and provided high resolution TEM analysis to show abundant prismatic dislocations in the as-fabricated nanostructured Ti6Al4V5Cu alloy (new Supplementary Fig. 8a-8d). We also compared the frequency of high Schmid factors between the scenarios when the hot deformation is dominated by basal slip (Supplementary Fig. 8e, 8g) and prismatic slip (Supplementary Fig. 8f, 8h). It is well recognized that plastic deformation usually starts in a grain with high Schmid factors. Comparing Fig. 8e with Fig. 8f (or Fig. 8g with Fig. 8h), it can be clearly seen that much higher frequency of high Schmid factors (0.3~0.5) occurs when prismatic slip is predominant. Taking both TEM and EBSD studies into consideration, the authors are confident that prismatic slip is the dominant deformation mechanism of the Ti6Al4V5Cu alloy. Details of changes in the revised manuscript are as follows:

Page 5, line 146- change "This is in agreement with our inverse pole figure (IPF) results (Fig. 2g). In the Ti6Al4V5Cu alloy, a strong prismatic texture which originated from the prismatic slip was characterized," to "This agrees with our TEM and electron backscatter diffraction (EBSD) analysis (Supplementary Fig. 8). In the Ti6Al4V5Cu alloy, a strong prismatic texture which originated from the prismatic slip was identified (Fig. 2g)."

Page 9, line 302- add "Schmid factor maps of the as-fabricated Ti6Al4V5Cu alloy were calculated with a hypothesis that basal slip or prismatic slip dominated the hot deformation." after "between α , Ti₂Cu and β phase."

Page 18, line 641- add a new Supplementary Fig. 8.

Supplementary Fig. 8 Identification of the dominant deformation mode during the formation of DPHS nanostructured Ti6Al4V5Cu alloy. **a**, High resolution TEM image in a local α grain with the zone axis of $\langle 0002 \rangle$. **b**, **c**, and **d**, Simulated lattice fringes of each prismatic plane in $\{11\bar{2}0\}$ family to show the edge component of $\langle a \rangle$ type dislocations lying in $\{11\bar{2}0\}$ prismatic planes. **e** and **f**, EBSD Schmid factor maps hypothesizing the hot deformation is dominated by basal slip and prismatic slip, respectively. **g** and **h**, frequency histogram of Schmid factors in **e** and **f** respectively.

(9) In figure 3c a legend explaining atom colours should be included.

Response:

We have followed reviewer's suggestion and added a legend explaining atom colours in Fig. 3c.

(10) The stages of deformation I, II, III and IV shown in figure 4f do not seem to be discussed explicitly anywhere in the text, figure captions or supplementary material. This needs to be addressed.

Response:

We have followed reviewer's suggestion and explained different stages of the superplastic deformation in the caption of Fig. 4 and in the main text. Details of changes are as follows:

Page 7, line 229- add “**details of the superplastic mechanism has been shown in Fig. 4f**” after “**should be governed by phase boundary sliding.**”

Page 7, line 241- change “**Schematic shown of the superplastic mechanism in the alloy.**” to “**Schematic flowchart of the superplastic mechanism in the Ti6Al4V5Cu alloy: I, Initial state; II, Phase boundary sliding in one tangential direction (yellow arrows); III, Phase boundary sliding in another equivalent tangential direction (purple arrows); IV, Alternating phase boundary sliding in different tangential directions sustains superplastic deformation along the loading direction, while Ti₂Cu/β shells suppress grain coarsening during the deformation.**”

(11) The standard of English and quality of writing are acceptable, although there are some rough edges. I recommend making the following stylistic changes:

everywhere in the text - replace "thermal-mechanical" with "thermomechanical"

everywhere in the text - replace "nano-lathy precursor" with something better. It is not quite English, certainly not proper scientific English.

everywhere in the text - What are transversal boundaries? DO the authors mean transverse boundaries?

page 1, line 16 - change "be applied" to "apply"

page 1, line 21 - change "Ti6Al4V5Cu" to "Ti6Al4V5Cu (wt.%)"

page 1, line 30 - change "has a great" to "has great"

page 1, line 30/31 - change "to be applied in" to "for application to"

page 2, line 44 - change "Over past" to "Over the past"

page 2, line 51 - delete "Basing on above theories", start sentence with "The stability"

page 2, line 63 - change "exert effective" to "exert an effective"

page 2, line 65 - change "that synchronises the thermodynamics and kinetics strategies" to ", which synchronises the thermodynamics and kinetics strategies,"

page 2, line 66 - change "alloy as a model alloy in" to "model alloy for"

page 2, line 73 - change "by room temperature" to "using room temperature"

page 2, line 82 - change "remained its" to "retained their"

page 2, line 84 - change "manufactured complex" to "manufactured into complex"

page 3, line 95 - change "should be originated from its" to "originate from the alloy's"

page 3, line 96 - change "microstructural architecture" to "microstructure" (no need to be too fancy)

page 3, line 96/97 - change "under the high-angle ... mode" to "using high-angle ... imaging"

page 3, line 98 - change "forming" to "form"

page 3, line 101 - change "This is why we call it" to "For these reasons, we have called it the"

page 3, line 102 - change "is credited to" to "is promoted by" or "is enabled by"

page 3, line 108 - change "The D step" to "The role of the D"

page 3, line 109 - change "Meanwhile, dual-phase" to "Meanwhile, the dual-phase"

page 3, line 122 - change "It well" to "This"

page 4, line 116 - change "short-listed as the" to "selected as the additional"

page 4, line 117 - delete "in this study"

page 4, line 135 - change "Williamson-Hall" to "the Williamson-Hall"

page 4, line 145 - change "dominated" to "dominant"

page 5, line 156 - change "The newly" to "While the newly"

page 5, line 156 - change "might be with" to "have"

page 5, line 157 - change "but they would" to "would be expected to"

page 5, line 164 - change "phase rapidly precipitated" to "phases rapidly precipitate"

page 5, line 165 - change "shells enveloping" to "shells that envelop"

page 5, line 167 - change "nanostructured stability" to "stability of the nanostructure"

page 5, line 177 - change "energy" to "energies"

page 5, line 179 - change "phase interfaces" to "types of interface"

page 5, line 185 - change "exert substantial" to "exert a substantial"

page 5, line 185 - change "grains hence" to "grains and hence"

page 5, line 186 - delete "Obviously" (dangerous word to use in scientific writing)

page 5, line 189 - change "we use atom probe tomography (APT) analysis" to "we used atom probe tomography (APT)"

page 6, line 194 - change "transferred" to "migrated" or "diffused"

page 6, line 201 - change "to hamper" to "to"

page 6, line 202 - change "grain inside" to "grain interiors"

page 6, line 203 - change "thus" to "and thus"

page 6, line 205 - change "due to" to "because the"

page 6, line 206 - change "easily get coarsened" to "readily coarsen"

page 7, line 218 - delete "a" before therm...

page 7, line 219 - change "conditions" to "condition"

page 7, line 221 - delete "observation"

page 7, line 226 - change "phase interface" to "interface"

page 7, line 230 - change "phenomenon has ever" to "has"

page 7, line 231 - change "dominated" to "dominant"

page 7, line 233 - change "would" to "are"

page 7, line 246 - change "applicable in" to "applicable to"

page 8, line 253 - delete "to bulks"

page 8, line 254 - change "through" to "using"

page 8, line 269 - change "as aforementioned" to "as the aforementioned"

page 8, line 282 - change "electrolytic polished by" a mix" to "electrolytically polished using"

page 9, line 291 - change "backscattered" to "backscatter"

page 9, line 291 - delete "analysis"

page 9, line 295 - change "data was analysed by" to "data were analysed using"

page 9, line 295 - change "code" to "coding"

page 9, line 297 - change "pointing in the direction parallel" to "relative"

page 9, line 299 - change "in bold lines" to "using bold lines"

page 9, line 300 - change "in thin lines" to "using thin lines"

page 9, line 303 - delete "observation"

page 9, line 306 - change "by a voltage" to "using a voltage"

page 9, line 309 - change "mode were" to "imaging was"

page 9, line 310 - change "analysis were" to "analysis was"

page 9, line 311 - change "in the shell" to "within the shell"

page 9, line 311 - change "mode was" to "imaging was"

page 9, line 313 - delete "analysis"

page 9, line 316 - change "Following, they were" to "They were subsequently"

page 9, line 321 - delete "observation"

page 9, line 323 - change "thick" to "thickness"

page 9, line 325 - change "drawn" to "milled"

page 9, line 325 - change "by the FIB" to "using the FIB"

page 9, line 327 - change "Markers were carved as light as possible" to "Markers were carved to the minimum possible depth"

page 9, line 327 - change "Dimensions" to "The dimensions"

page 9, line 327 - change "markers was" to "markers were"

page 10, line 337 - change "In terms of" to "For an"

page 10, line 349 - change to "Using Density Functional Theory first principles calculations, we have ..."

Response:

The authors sincerely appreciate the reviewer's kind advice and extreme patience in picking up these errors. All the above suggestions have been adopted in our revised manuscript.

Reviewer #2

(1) A major concern is about the explanation for the formation of the dual-phase honeycomb shell nanostructure in the designed Ti6Al4V5Cu alloy during hot deformation (Fig. 3). The results showed that the addition of 5%Cu in a conventional Ti6Al4V alloy has enabled the

formation of finer martensitic laths after quenching from the beta phase and changed the deformation mechanisms and microstructural evolution during subsequent hot compression at 1023 K. It is no doubt that the development of the nanoscale alpha grains with the dual-phase shell structure during the hot compression is related to the combined effect of dynamic recovery and recrystallization with strain assisted phase transformation from alpha prime into alpha +beta+Ti₂Cu. However, the authors speculated a three-step process for the microstructural evolution during hot compression: (1) Transverse boundaries that are perpendicular to the lath boundaries are introduced by the activation of prismatic slip, resulting in the formation of equiaxed alpha prime grains with stable low angle boundaries at the beginning of hot deformation; (2) the low angle boundaries soon evolve into high angle boundaries through dynamic recovery and dynamic recrystallization; (3) conjugated β and Ti₂Cu phases rapidly precipitate at the HAGBs, forming “protective” dual-phase shells enveloping the nano-grains. In the section of Methods, the heating rate to heat the sample to the hot compression temperature and the isothermal holding time before compression were not described. Therefore, it is not known whether the transformation of alpha prime to alpha phase already took place before compression. More importantly, no observation of the microstructure at lower compression strains was provided to support the speculation about the evolution from low angle to high angle grain boundaries before the precipitation of beta and Ti₂Cu at grain boundaries. A detailed electron microscopy study is needed before a conclusion can be reached about the formation mechanism of the dual-phase honeycomb shell nanostructure. In this manuscript, if further evidence cannot be provided, it is suggested to simplify the related description and discussion and leave a more detailed electron microscopy investigation to reveal the formation mechanism for a future study.

Response:

The authors appreciate the reviewer’s valuable suggestion and have supplemented hot compression tests at height reduction of 20%, 40%, 60%, 80%, respectively at 1013 K with a compression strain rate of 2 s⁻¹. The microstructure at different stages was characterized by TEM/EDS to clarify the formation mechanism of the dual-phase honeycomb shell nanostructure (Supplementary Fig. 9). When the compression strain increases from 20% to 80%, the extent of Cu and V segregation gradually increases based on the EDS mapping results. It appears that Ti₂Cu

and β phase did not precipitate from matrix until the compression ratio approaches 60% though the authors cannot fully rule out the possibility of formation of Ti_2Cu and/or β embryo in the early stage of hot compression, which might be beyond the resolution of EDS mapping. Nevertheless, Supplementary Fig. 9 manifests clearly that the phase transformation of $\alpha' \rightarrow \alpha + \beta + Ti_2Cu$ did not take place prior to hot compression.

In addition, the supplemented TEM results provide solid evidence to support the three-step process of the microstructural evolution during hot compression. Step I (Supplementary Fig. 9b): when the compression strain was 20%, numerous transverse boundaries inside α' lath were observed. The formation of these transverse boundaries was attributed to the prismatic slip (justification of the slip mode is provided in new Supplementary Fig. 8 to respond to the major concern of the other reviewer). Step II (Supplementary Fig. 9c): when the compression strain was 40%, dynamic recovery (DRV) and dynamic recrystallization (DRX) were observed in some areas, which supports the description in the initial manuscript that “the low angle boundaries soon evolve into high angle boundaries through dynamic recovery and dynamic recrystallization”. Step III (Supplementary Fig. 9d, 9e): when the compression strain was 60%~80%, β and Ti_2Cu phases started precipitating along α grain boundaries, which supports the description in the initial manuscript that “conjugated β and Ti_2Cu phases rapidly precipitate at the HAGBs, forming ‘protective’ dual-phase shells enveloping the nano-grains”.

Details of changes in the revised manuscript are as follows:

Page 5, line 148- add “(Supplementary Fig. 9b)” after “hence transverse boundaries formed”.

Page 5, line 158- add “(Supplementary Fig. 9c)” after “soon evolve into HAGBs through dynamic recovery and dynamic recrystallization”.

Page 5, line 162- change “(Supplementary Fig. 7)” to “(Supplementary Fig. 9d, 9e, 10)”

Page 8, line 260- change “They were isothermally compressed at 1013 K” to “They were heated to 1013 K at a heating rate of 2~5 K/s, then held for 10 minutes and isothermally compressed”.

Page 8, line 263- add “To clarify the formation mechanism of the DPHS nanostructure during hot deformation, additional hot compression tests at height reduction of 20%, 40%, 60%, 80% were also performed.” After “with grinding machines by 2 mm.”

Page 19, line 642- add a new Supplementary Fig. 9.

Supplementary Fig. 9 TEM/EDS analysis of the microstructure at different stages during hot compression. **a**, Before hot compression. **b**, At 20% compression strain, transverse boundaries were introduced through prismatic slip, and equiaxed α' grains with low angle grain boundaries (LAGBs) formed. **c**, At 40% compression strain, dynamic recovery (DRV) and dynamic recrystallization (DRX) took place, and LAGBs evolved into high angle grain boundaries (HAGBs). **d, e**, At 60%~80% compression strain, conjugated β and Ti_2Cu phases massively precipitated along the HAGBs, forming the *DPHS* nanostructure.

(2) Fig. 3d shows pole figures of alpha, beta and Ti₂Cu phases. This indicates that the EBSD applied in this work allowed the detection of EBSD patterns from the thin layers of individual beta and Ti₂Cu phases. However, the EBSD orientation maps shown in Fig. 3a (also Fig. 2g) show only alpha grains without a shell structure, as revealed in the TEM images (Fig. 2a). How have the EBSD maps in Fig. 3a and Fig. 2g been processed? Have any data cleaning procedures been applied? If so, the EBSD data processing needs to be described in the Methods. Also, the sample coordinate system should be indicated in the pole figures in Fig. 3d.

Response:

The first map in Fig. 3a is taken as an example here to explain how the EBSD data was processed. Fig. R1 shows the phase distribution maps from the raw data, where both β and Ti₂Cu phases were detected and the quantity of counts were summarized in Table R2 though their volume fraction is very low (0.54% for β phase and 0.82% for Ti₂Cu phase, respectively). It is expected that many β and Ti₂Cu precipitates that are thinner than the step size of EBSD mapping (20 nm) cannot be indexed and contributed to the “zero solutions” (19.84%). That explains why both β and Ti₂Cu phase are scattered points in the phase distribution map of Fig. R1, rather than continuous shells around α grains shown in TEM images.

After raw data collection, automatic noise reduction using the Chanel 5 software was performed to reduce the zero solutions then overlaid with grain boundaries, as shown in Fig. R2. Note that the main purpose of Fig. 3a is to show the α grain size stability at different elevated temperatures. So, the IPF maps of α phase solely appear more succinct than superimposing all three phases together. In contrast, the main purpose of Fig. 3d is to identify energetically favorable orientation relationship among these three phases. So, the pole figures of principle planes/directions in all three phases are shown in Fig. 3d.

Details of changes in the revised manuscript are as follows:

Page 6, line 207- add the compression direction (CD) in the pole figures of Fig. 3d.

Page 6, line 210 (In the Caption of Fig. 3)- add “Only the IPF maps of α phase is shown for brevity” after “a, IPF maps of the Ti₆Al₄V₅Cu sample after annealed at various temperatures”.

Page 9, line 295- change “Acquired data was analyzed by HKL-Channel 5 software.” to “Acquired data were analyzed using HKL-Channel 5 software after automatic noise reduction process.”

Fig. R1 Raw EBSD data of phase distribution maps. **a**, All phases distribution. **b**, α phase distribution. **c**, β phase distribution. **d**, Ti_2Cu phase distribution. β and Ti_2Cu phases are scattered points distributed along α grain boundaries.

Table R2 Statistic results of EBSD raw data

Name	Counts	Volume fraction/%
α	68995	78.80
β	470	0.54
Ti_2Cu	721	0.82

Fig. R2 Data processing procedures for IPF maps in Fig. 3a.

(3) In situ SEM observations have shown clearly that phase boundary sliding associated with the fine structure is the dominant deformation mechanism responsible for the excellent superplasticity. Microstructural observation in the sample deformed to 40% showed no grain growth. However, microstructural observations at higher strains should be presented to verify whether grain growth has occurred during the superplastic deformation. In conventional fine-grained superplastic materials, it has been well-established that gradual grain growth occurs during superplastic deformation, which is a mechanism for hardening that is considered to be helpful in addition to the strain rate hardening to maintain a smooth necking of the tensile specimen during superplastic deformation. The grain growth and its role in superplastic deformation in the present alloy cannot be ruled out before experimental evidence is provided.

Response:

The authors completely agree with the reviewer's comment, and had planned to investigate the microstructure evolution at higher strains through in-situ SEM observations. However, the tensile strain was limited by both the maximum displacement of the micro-tensile testing sample stage and the size of built-in heating device (Fig. R3). That is why the maximum tensile strain was 40% in the initial manuscript. Alternatively, we have performed post deformation TEM observation after tensile test at 923 K. As shown in Supplementary Fig. 4b, no noticeable grain coarsening was observed even after tensile strain exceeds to 1000%. This indicates that the contribution of the grain growth induced strengthening is negligible in this nanostructured Ti6Al4V5Cu alloy in this tensile testing condition. Nevertheless, it is expected that noticeable grain growth will occur inevitably when the tensile test is conducted at higher temperatures (> 973 K). That will be definitely an interesting research topic, and we will make further investigations in the future.

Fig. R3 The micro-tensile testing sample stage for the in-situ SEM observation

Supplementary Fig. 4 TEM observation of the nanostructured Ti6Al4V5Cu alloy. **a**, After room temperature tensile test. **b**, After 923 K tensile test with a deformation strain of exceeding 1000%. Images were taken under bright field (BF), high angle annular dark field (HAADF), and X-ray energy dispersive spectroscopy (XEDS) mode. The nanostructure was barely evolved after room and high temperature tensile tests, indicating the material possessed excellent mechanical and thermomechanical coupling stability.

(4) At last but not least, there are quite a few grammatical and typographic errors in the main text and in figure captions. A careful check of the language is required. Examples of errors are:

Line 92: Strength versus stain curves (Stress versus strain curves)

Line 122 and Fig. 4f: Schematic shown..... (Schematic showing.....)

Line 138: lathy variants (lath variants)

Line 260: rolling bars (forging bars)

Lines 278 and 279: strain rale (strain rate)

Line 293: vibration polished (vibration polishing)

Response:

The authors sincerely appreciate the reviewer's corrections of these errors, all of which have been amended in our revised manuscript.

REVIEWERS' COMMENTS

Reviewer #1 (Remarks to the Author):

I have gone through the revised manuscript and through the authors responses to the reviewers' queries. The authors have addressed all of my original concerns with the manuscript and have carried out additional experiments and simulations, which has improved the overall quality of the work and helped validate the authors' findings. Given the novelty of the work and the overall level and quality of scientific effort put in, I am happy to recommend the article for publication.

Reviewer #2 (Remarks to the Author):

I am glad to see the authors have rather adequately addressed the comments of reviewers and the manuscript has been improved to be acceptable.

I have noticed several typos or errors during the review process, and suggest making the following corrections:

Line 147 : change “resulted in the lath width significant decreasing from” to “resulted in a significant decrease in the lath width from”

Line 222 and line 225: change “after annealed” to “annealed” or “after annealing”

Line 279: change “the forging-ending temperature” to “the finish forging temperature”

Line 284: change “The as-prepared Ti6Al4V5Cu material were plates with the 285 diameter of (175±10) mm.” to “The as-compressed Ti6Al4V5Cu samples were round plates with a diameter of 175±10 mm.”

Line 303: change “strain rale” to “strain rate”

Line 325: change “bold line” to “bold lines”

Line 327: change “thin line” to “thin lines”

Line 347: change “tip diameter” to “a tip diameter”

Line 350: change “reconstruction were” to “reconstruction was”

Line 367: change “basing on” to “based on”

Line 380: change “all above” to “all the above”

** See Nature Research's author and referees' website at www.nature.com/authors for information about policies, services and author benefits

Point by point response to reviewer comments

The authors sincerely appreciate reviewers' pertinent comments on our manuscript entitled "Manufacture-friendly nanostructured metals stabilized by dual-phase honeycomb shell" (NCOMMS-21-19879A). These comments are highly valuable to improve the quality of our manuscript. Amended parts have been highlighted in the revised manuscript. The point-by-point responses to reviewers' comments are as follows:

Reviewer #1 (Remarks to the Author):

I have gone through the revised manuscript and through the authors responses to the reviewers' queries. The authors have addressed all of my original concerns with the manuscript and have carried out additional experiments and simulations, which has improved the overall quality of the work and helped validate the authors' findings. Given the novelty of the work and the overall level and quality of scientific effort put in, I am happy to recommend the article for publication.

Reviewer #2 (Remarks to the Author):

I am glad to see the authors have rather adequately addressed the comments of reviewers and the manuscript has been improved to be acceptable.

I have noticed several typos or errors during the review process, and suggest making the following corrections:

Line 147: change "resulted in the lath width significant decreasing from" to "resulted in a significant decrease in the lath width from"

Line 222 and line 225: change "after annealed" to "annealed" or "after annealing"

Line 279: change "the forging-ending temperature" to "the finish forging temperature"

Line 284: change "The as-prepared Ti6Al4V5Cu material were plates with the 285 diameter of (175±10) mm." to "The as-compressed Ti6Al4V5Cu samples were round plates with a diameter of 175±10 mm."

Line 303: change "strain rale" to "strain rate"

Line 325: change "bold line" to "bold lines"

Line 327: change "thin line" to "thin lines"

Line 347: change "tip diameter" to "a tip diameter"

Line 350: change "reconstruction were" to "reconstruction was"

Line 367: change "basing on" to "based on"

Line 380: change "all above" to "all the above"

Response:

The authors sincerely appreciate the reviewer's kind advice and extreme patience in picking up these errors. All the above suggestions have been adopted in our revised manuscript.